

# Habitual and ready positions in female table tennis players and their relation to the prevalence of back pain

Ziemowit Bańkosz[1,*] and Katarzyna Barczyk-Pawelec[2,*]

[1] Faculty of Sports, University School of Physical Education in Wrocław, Wrocław, Poland
[2] Faculty of Physiotherapy, University School of Physical Education in Wrocław, Wrocław, Poland
* These authors contributed equally to this work.

## ABSTRACT

**Background:** The current body of knowledge shows that there is very little research into the occurrence and scale of asymmetry or postural defects in table tennis. It is interesting which regions of the spine are exposed to the greatest changes in the shape of its curvatures and whether the asymmetrical position of the shoulder and pelvic girdles in table tennis players changes when adopting the ready position. Consequently, can overload occur in certain parts of the spine and can the asymmetry deepen as a response of adopting this position? The reply to these questions may be an indication of the need for appropriate compensatory or corrective measures. Therefore, the aim of the study was to evaluate the effect of body position during play on the change in the shape of anterior–posterior spinal curvatures and trunk asymmetry in table tennis players.

**Methods:** To evaluate body posture the photogrammetric method based on the Moiré phenomenon with equipment by CQ electronic was applied. The study involved 22 female players practicing competitive table tennis (the age of 17 ± 4.5, with the average training experience of 7 ± 4.3 years, body mass of 47.8 ± 15.8, and body height of 161.2 ± 10.4). Each participant completed an author's own questionnaire on spinal pain. The shape of curvatures in the sagittal and frontal plane was evaluated in the participant in the habitual standing position and in the table tennis ready position. Descriptive statistical analysis was performed and the significance of differences was tested using the Mann–Whitney $U$ test.

**Results and Conclusions:** This study demonstrated the dominance of kyphotic body posture in table tennis players, which can be caused by many hours of using the ready position during playing. After adopting this position, there are significant differences in the angles of anterior and posterior spinal curvatures compared to the habitual posture. This may be the cause of overloads and pain complaints reported by the study participants. Adopting the ready position is also associated with an increase in asymmetry in the position (rotation) of the pelvis and spinous processes (frontal plane). Therefore, training programs should be extended with exercises that relieve the spine in the vertical line and exercises that improve symmetry of the work of the upper limbs, body trunk muscles and the pelvis.

Corresponding author
Ziemowit Bańkosz,
ziemowit.bankosz@awf.wroc.pl

## INTRODUCTION

Table tennis is one of the fastest sports (*Kondric, Zagatto & Sekulic, 2013*). This is mainly due to the short distance between the players (table length is 2.74 m) and the speed of the flying ball (up to about 40 m/s). For this reason, the players have very little time to react, ranging from 0.2 to 0.4 s. Except for the service, all players' actions represent the response to the opponent's play. Therefore, each player has to evaluate the parameters of the flying ball: where and how the ball will bounce on the table and what the speed and rotation will be. Then the player must precisely choose the parameters of the stroke such as its type, strength and direction, angle of the racket, place of hitting the ball and adopt the right position to perform the play. All this causes the player to act in a constant shortage of time. It is therefore essential to remain ready. This is expressed by taking and maintaining the so-called ready position, in which the lower and upper limbs are flexed, the torso is significantly leaned forward, the rocket is kept in front of the player's body, the center of gravity of the body is shifted forward, the body weight is kept on the forefoot, etc. (Fig. 1; *Hudetz, 2005*).

Another characteristic element of table tennis is the one-sidedness and asymmetry of muscle work because the player plays with one hand. The impact movements are therefore asymmetrical and significantly load one side of the body. Impact movements are characterized by high speed and the impact force is generated based on the principle of "proximal to distal sequence," using the work of the whole body (*Iino & Kojima, 2009*; *Bańkosz & Winiarski, 2017*, *2018*). The movements of the pelvic girdle, torso and shoulder girdles (*Iino & Kojima, 2009*, *2011*, *2016*; *Bańkosz & Winiarski, 2018*), especially in the transverse and frontal plane, are of great importance to the achievement of a high impact force.

Postural defects (excessive spine curvatures, scoliosis), limb distortions and asymmetry of body build are the factors leading to pain syndromes, degenerative states, disorders of motor functions or even internal organs functioning (*Zeyland-Malawka & Prętkiewicz-Abacjew, 2006*). Researchers dealing with body posture, the symmetry of body build or proportions of athletes' bodies often find that the risk of occurrence of excessive morphological asymmetry or spinal pain syndromes in athletes is high. *Hobbs et al. (2014)* have identified a high risk of chronic spinal pain and morphological asymmetry in female and male equestrian athletes. The study found a high correlation between the incidence of injury and certain body mechanics disorders in football players (*Watson, 1995*). Increased lumbar lordosis and increased or decreased distance between the knees were often associated with muscle strain, while increased thoracic kyphosis and shoulder and trunk asymmetry were associated with back pain. The risk of injury is also high as a result of functional asymmetry, which was found in soccer players (*Read et al., 2017*). *Toraman & Yaman (2001)* demonstrated the relationship between the occurrence of asymmetry in different parts of the body and the occurrence of injuries in adolescents. *Krzykała et al. (2018)*, who examined asymmetry in hockey players, emphasized the role of monitoring of the magnitude of the asymmetry in preventing injuries and health problems linked to morphological asymmetry. *Morton & Callister (2010)* found a frequent

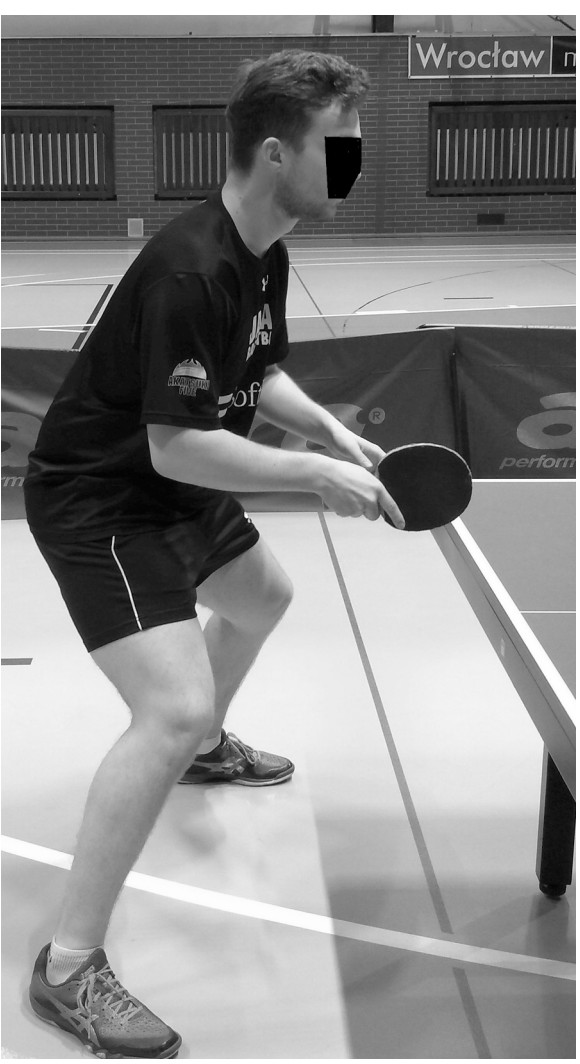

**Figure 1 Ready position.**               

occurrence of transient abdominal pain after exercise in cases of increased thoracic kyphosis or lumbar lordosis.

Many researchers have pointed to the occurrence of asymmetry in athletes. *Grabara & Hadzik (2009)*, assessing the body build of young female and male athletes, found numerous asymmetries with respect to waist triangles and shoulder blade position and tendencies for increased thoracic kyphosis.

*Grabara (2018)* found that thoracic kyphosis increased while lumbar lordosis decreased in young handball players during a 2-year training period. A large number of pelvic asymmetries in athletes practicing sports with one-sided domination (limb use, rotating upper body) was stressed by *Bussey (2010)*, who examined hockey players, field hockey players and speed skaters.

However, some researchers point out that practicing sport involves correcting or symmetrical development of body posture. Such results have been documented by
researchers in the field of taekwondo (*Wąsik et al., 2015*), gymnastics (*Radaŝ & Trost Bobiĉ, 2011*), or karate (*Drzał-Grabiec & Truszczyńska, 2014*). *Maloney (2019)*, in a review of available studies, pointed out that there is no convincing evidence of asymmetry in athletes and that it is sporting activity that can counteract such asymmetries.

The current body of knowledge shows that there is very little research into the occurrence and scale of asymmetry or postural defects in table tennis. It is interesting to see how and to what extent the body posture changes during the adoption of a typical playing ready position. To be more specific, the question is which regions of the spine are exposed to the greatest changes in the shape of its curvatures and whether the asymmetrical position of the shoulder and pelvic girdles in table tennis players changes when adopting the ready position? It is also interesting if table tennis players declare the occurrence of back pain, what are the scale and consequences of this pain and if this pain occurrence is correlated to any of measured body posture parameter? There is no data in the literature concerning this problem. The reply to these questions may be an indication of the need for appropriate compensatory or corrective measures (*O'Sullivan et al., 2002*; *Żuk et al., 2019*). Therefore, the aims of the study were to: evaluate the effect of body position during play on the change in the shape of anterior–posterior spinal curvatures and trunk asymmetry in table tennis players and to establish the correlation between prevalence of back pain and parameters of body posture in table tennis players.

## MATERIALS AND METHODS

### Participants

The method of sampling in the research was judgmental sampling—the research concerned female table tennis players who have been practicing table tennis more than 2 years. The research involved 22 female players practicing competitive table tennis at the age of 17 ± 4.5, with the average training experience of 7 ± 4.3 years, body weight of 47.8 ± 15.8 and body height of 161.2 ± 10.4. The research was done during afternoon session of training, between 5.00 and 8.00 PM. All participants trained at least 2.5 h a day 6 times a week, and some of them more often (twice a day). All of them were informed about the research aim and procedures and signed informed consent to participate in the experiment. The research was approved by The Senate's Research Bioethics Commission at the University School of Physical Education in Wrocław. After signing the consent, each participant completed an author's own questionnaire on spinal pain, in which they answered the following questions: (1) How often do you complain about back pain? (never or almost never; rarely; occasionally; often; very often). (2) Which sections of your back do you find to be the most often painful? (I have no back pain; lumbar spine; thoracic spine; cervical spine; all regions). (3) What is the most frequent pain intensity on a scale from 0 to 10? (0—no pain to 10—unbearable pain). (4) Have you ever had to give up training (competition) because of back pain? (yes, no) (5) Is the spinal pain getting worse? (during training; immediately after the training session, sometime after the training session; no pain). (6) If pain occurs, what is its nature? (radiating to the lower limb; radiating to the upper limb, local without radiation). The answers for the question

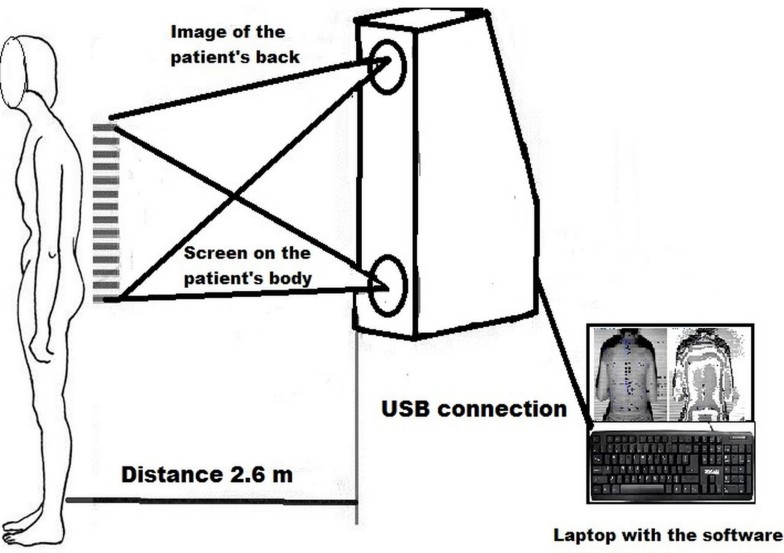

**Figure 2 Scheme of the research stand.**

number one helped to divide participants into two groups. Group 1 (8 players) gave answers: never, almost never or rarely and group 2 (14 players) gave answers: occasionally; often or very often.

## Procedures

Body posture assessment in all patients was performed with a device for computer analysis of the shape of anterior–posterior curvatures of the spine and trunk asymmetry using the photogrammetric method and a fourth-generation moiré apparatus (CQ Elektronik System, Wroclaw, Poland) that maps the anteroposterior spine curvature (*Porto et al., 2010*; *Barczyk-Pawelec & Sipko, 2017*) (Fig. 2). The moiré technique is based on a type of optical distortion created by the interference of light waves, as if an image was being refracted. A series of visible lines are projected on the surface of the back, which at different angles are distorted depending on the distance of a given anatomical marker from the projector. In effect, this photogrammetric method mirrors the shape of the back (Fig. 3).

Before the examination, the following points were marked on the body of the participant with a washable black marker: spinous processes of spine vertebrae from C7 to S1 and thoracic–lumbar transition, acromions, inferior angles of scapulae and posterior superior iliac spine. All determinations on the body were made by the same physiotherapist experienced in this type of examination. Based on the contour of the curvature of the spine, the program automatically determined the peaks of thoracic kyphosis and lumbar lordosis. Three-dimensional coordinates of body surface were obtained based on the recorded images of the body trunk of the participants. The parameters determining the anterior–posterior spinal curvatures, the sagittal inclination of the trunk, magnitude of asymmetry within the shoulder and pelvic girdles, and trunk inclination in the frontal plane were calculated.

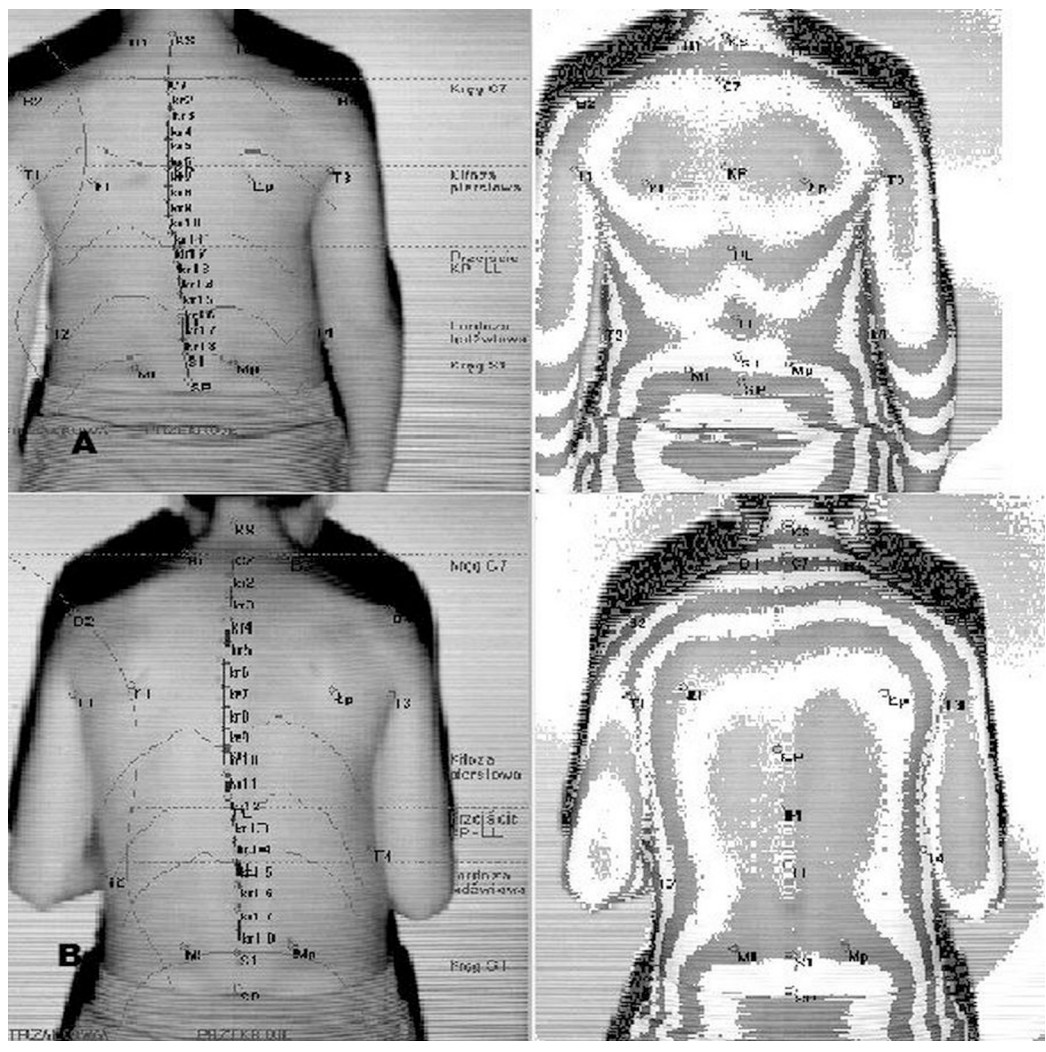

**Figure 3 Body posture examination using the photogrammetric method in habitual posture (A) and in the ready position (B).**

In the sagittal plane, the following angular parameters of spinal curvatures were evaluated and analyzed (Fig. 3):

– angle of inclination of the lumbosacral spine ($\alpha$),

– angle of inclination of the thoracolumbar spine ($\beta$),

– angle of inclination of the upper part of the thoracic region ($\gamma$),

– angle of sagittal inclination of the trunk (KPT). Negative angles indicate the forward inclination of the trunk relative to the vertical line,

– angle of thoracic kyphosis (KKP),

– angle of lumbar lordosis (KLL),

– depth of thoracic kyphosis (GKP),

– depth of lumbar lordosis (GLL).

The following body asymmetries were evaluated and analyzed in the frontal plane:
(a) Angular parameters (expressed in degrees):

- KNT—angle of trunk inclination,
- KLB—angle of shoulder line inclination,
- KNM—pelvic inclination angle,
- KSM—pelvic rotation angle.

(b) Length and depth parameters (expressed in mm):

- UL—difference in the positions of the inferior angles of scapula,
- OL—difference in the distance of inferior angles of scapula from the spine,
- TT—difference in the height of the waist triangles,
- TS—difference in the width of the waist triangles,
- UK—deviation of spinous processes from the line of the spine.

The magnitude of asymmetry was established on the basis of differences in the placement of osteal points within the trunk. Intervals of these differences were determined for angular and length parameters, distinguishing three levels of asymmetry according to *Bibrowicz (1995)*.

For the angle indices (KNT, KLB, KNM, KSM), it was assumed that:

- difference of $0° < x \leq 1.5°$ means no asymmetry,
- difference of $1.5° < x < 3°$ means moderate asymmetry,
- difference of $x \geq 3°$ indicates severe asymmetry.

For linear asymmetry indices (UL, OL, TT, TS, UK), it has been assumed that:

- difference of $0 < x \leq 5$ mm means no asymmetry,
- difference of $5 < x < 10$ mm means moderate asymmetry,
- difference of $x \geq 10$ mm means severe asymmetry.

Body posture was classified on the angular values of the anteroposterior spinal curvatures (compensation index) using the formula $\mu = \gamma - \alpha$, in which $\mu$ was defined as three possible body posture types. The first was a kyphotic-type posture (KT), featuring excess thoracic kyphosis compared with lumbar lordosis in which $\mu > 3$, $\gamma + \beta \geq 29°$ and $\alpha + \beta < 25°$; the second a lordotic'type posture (LT) whereby lumbar lordosis exceeded thoracic kyphosis and $\mu < -3$, $\alpha + \beta \geq 25°$ and $\gamma + \beta < 29°$; and the third was the balanced type (BT) with approximately equal curvatures as defined by $-3 \leq \mu \leq 3$ and $33° < \alpha + \beta + \gamma$ (*Zeylan-Malawka, 1999*; *Barczyk-Pawelec & Sipko, 2017*). The shape of curvatures in the sagittal plane was evaluated in the participant in the habitual standing position and in the table tennis ready position after a verbal instruction: "Adopt the ready position!" without giving any additional instructions or guidelines on the quality of the new position. The only thing that the participant could not do was crossing the line determining the distance between the camera and the participant with his or her heels.

First of all, body posture was assessed in a habitual standing position without shoes. The test participant stood in a habitual standing position within the field of vision of the camera at a distance of 2.6 m. The participant's feet were positioned on a line parallel to the measurement stand, spaced at the width of the hips. The knee joints were extended and the body weight was evenly distributed on both lower limbs. The upper limbs were placed freely along the torso, the head was positioned freely, and the eyes were looking ahead. After recording the shape of the upper body in the habitual standing position, the examined person, on the instruction of "Adopt the ready position!," adopted the given position and after 5 s, another image of the back was recorded.

## Statistics

The parameters obtained from the examinations were subjected to statistical analysis. Descriptive statistical analysis was performed (normality of distribution was tested by means of the Shapiro–Wilk test). Means, standard deviations and confidence intervals for mean CI 95% were calculated for all measured parameters. The significance of differences between habitual and ready positions was tested using the Mann–Whitney $U$ test with the level of statistical significance set at $p \le 0.05$, and $d$-Cohen's effect was calculated. The $U$ Mann–Whitney test was also used to examine differences between body posture parameters of group 1 and 2. This helped to assess the relation between the parameters of body posture and the frequency of occurrence of pain declared by participants. Statistica 10 package (Statsoft Inc., Tulsa, OK, USA) was used for calculations.

## RESULTS

A survey conducted immediately before the examination concerning the incidence of back pain showed that five athletes never (or almost never) complained about back pain. Three people complained rarely, 7—occasionally, 3—often and 4—very often. The thoracic region of the spine was considered to be the most common painful regions of the spine (11 people), followed by the lumbar region (eight people) and cervical region (two people). Four people declared no pain symptoms. The most frequent pain intensity indicated by the respondents (on a scale from 0—no pain to 10—unbearable pain) was five (six people), followed by six (six people), four (two people) and seven (two people). One person reported the intensity of eight, whereas four people—the intensity of zero. Eight people declared that due to spinal pain they had to stop training (or competition). In 10 people, the pain increased during training, in 4—immediately after training and in 3—some time after the training session. In most of the respondents, pain was local, without radiation (13 people). Four people reported pain radiating to the upper limb.

In the sagittal plane, the free posture of table tennis players was characterized by slightly deepened thoracic kyphosis, especially in the upper part. Based on the compensation index, a kyphotic type (KT) of posture was found. The depth of thoracic kyphosis (GKP) was also higher than that of lumbar lordosis (GLL). In the frontal plane, table tennis players were characterized by significant asymmetry, exceeding 10 mm, within the parameters of the difference in height and width of waist triangles (TT and TS) and the difference in the distance between lower shoulder blade angles and spine (OL).

**Table 1 Results of examinations in a group of players in the habitual standing position and the ready position: means, standard deviations (SD) and confidence intervals (CI 95%), p-values of the Mann–Whitney U-test and d-Cohen's values.**

| Table tennis players ($n = 22$) | Mean ± SD (CI 95%) | | p-Values of the Mann–Whitney U-test | d-Cohen's |
|---|---|---|---|---|
| | Habitual position | Ready position | | |
| α [deg] | 10.45 ± 5.54 [8.00–12.90] | 34.70 ± 15.76 [27.71–41.68] | **<0.01** | **1.43**** |
| β [deg] | 7.43 ± 4.31 [5.52–9.34] | 20.58 ± 9.18 [16.51–24.65] | **<0.01** | **1.35**** |
| γ [deg] | 13.92 ± 5.27 [11.59–16.26] | 43.24 ± 7.97 [39.71–46.77] | **<0.01** | **1.80**** |
| CI [deg] | 3.47 ± [8.34 [−0.22–7.17] | 13.73 ± 16.73 [6.31–21.14] | **0.01** | **0.73*** |
| KPT [deg] | −2.98 ± 3.87 [−4.69 to −1.26] | −29.94 ± 12.11 [−35.31 to −24.57] | **<0.01** | **−1.66**** |
| GKP [deg] | 13.63 ± 8.39 [9.91–17.35] | −38.98 ± 24.85 [−50.00 to −27.96] | **<0.01** | **−1.63**** |
| GLL [deg] | −12.33 ± 9.14 [−16.39 to −8.28] | 21.89 ± 15.65 [14.95–28.83] | **<0.01** | **1.60**** |
| KNT [deg] | 1.50 ± 0.94 [1.08–1.91] | 2.21 ± 2.06 [1.29–3.12] | 0.50 | 0.44 |
| KLB [deg] | 1.19 ± 0.88 [0.80–1.58] | 1.31 ± 1.17 [0.79–1.83] | 0.99 | 0.12 |
| UL [mm] | 2.27 ± 1.59 [1.56–1.59] | 1.48 ± 1.14 [0.98–1.99] | 0.09 | **−0.55*** |
| OL [mm] | 10.85 ± 8.63 [7.02–14.68] | 9.26 ± 9.65 [4.99–13.54] | 0.27 | −0.17 |
| TT [mm] | 11.93 ± 9.05 [7.92–15.94] | 14.57 ± 10.77 [9.79–19.34] | 0.49 | 0.27 |
| TSm [mm] | 9.26 ± 8.63 [5.43–13.08] | 11.65 ± 12.50 [6.11–17.19] | 0.34 | 0.22 |
| KNM [deg] | 1.51 ± 1.71 [0.75–2.27] | 1.38 ± 1.27 [0.81–1.94] | 0.97 | −0.09 |
| KSM [deg] | 4.21 ± 2.77 [2.98–5.44] | 10.30 ± 14.30 [3.96–16.63] | 0.34 | **0.57*** |
| UK [mm] | 4.30 ± 2.64 [3.13–5.48] | 6.24 ± 3.69 [4.61–7.88] | 0.06 | **0.58*** |

Note:
α, angle of inclination of the lumbosacral spine; β, angle of inclination of the thoracolumbar spine; γ, angle of inclination of the upper part of the thoracic region; CI, compensation index; KPT, angle of sagittal inclination of the trunk; KKP, angle of thoracic kyphosis; KLL, angle of lumbar lordosis; GKP, depth of thoracic kyphosis; GLL, depth of lumbar lordosis; KNT, angle of trunk inclination; KLB, angle of shoulder line inclination; KNM, pelvic inclination angle; KSM, pelvic rotation angle; UL, difference in the positions of the inferior angles of scapula; OL, difference in the distance of inferior angles of scapulae from the spine; TT, difference in the height of the waist triangles; TS, difference in the width of the waist triangles; UK, deviation of spinous processes from the line of the spine. Bold values mean significant difference and significant effect size. Differences are significant when $p \leq 0.05$. Effect size is medium when Cohen's d is $0.5 \leq 0.8$ (*) and large when $d > 0.8$ (**).

Furthermore, table tennis players also showed asymmetry at the pelvic rotation angle in the transverse plane (KSM). The remaining analyzed parameters for angular and linear asymmetries (KNT, KLB, UL, KNM, UK) were at a moderate level (difference of 1.5–3° in case of angles, difference of 5–10 mm in case of linear measure—see Table 1).

The change in body position has a significant effect on the angles of anterior–posterior spinal curvatures, the angle of trunk inclination and the depth of thoracic kyphosis and lumbar lordosis (Table 1). Significant changes in all three angles of spinal curvatures were observed between the habitual standing position and the ready position. After changing the position from habitual posture to the ready position, the angle of inclination of the upper thoracic and lumbosacral regions increased significantly: their values tripled and the angle of the thoracolumbar region increased more than twice. The value of the angle of trunk inclination (KPT) increased tenfold ($p < 0.01$). Furthermore, as a result of adopting the ready position, significant changes in the depth of both thoracic kyphosis and lumbar lordosis were observed. In these cases, the effect size calculated based on the d-Cohen test was very high ($d$-Cohen $\geq 1.0$). The change in body position had only a slight effect on the change in the magnitude of asymmetry in body trunk and concerns mainly pelvic rotation (KSM) and the deviation of the spinous processes (UK). The medium effect size ($d$-Cohen $\geq 0.5$) was found in this case.

Test *U* showed also the difference between group 1 and 2 of participants in the case of years of experience—group 1: 3.3 ± 1.7 *y* and group 2: 9.2 ± 3.9 *y*, with *p* < 0.01. The significant difference was also indicated in the case of angle β in habitual position (*p* = 0.03).

## DISCUSSION

The aims of the study were to: evaluate the effect of body position during play on the change in the shape of anterior–posterior spinal curvatures and trunk asymmetry in table tennis players and to establish the frequency of occurrence of back pain and its correlation to body posture parameters as well as the scale of back pain in table tennis players. Few scientific studies have analyzed the parts of the spine that undergo the greatest changes and their direction. This is of great cognitive importance because table tennis players adopt a specific body position for a long period of time during many hours of training and during the game. Forced positions, that is, flexion-based posture adopted during readiness for play, can lead to overloading of the lumbar spine.

Kyphotic posture, which was found in our study in table tennis players, may cause various physiological and functional disorders of the player's musculoskeletal system. According to *Nachemson (1987)*, the greatest pressure on the intervertebral discs in the lumbar region (mainly on the 3rd lumbar disc) is observed in the standing position with a simultaneous inclination of the body towards the front, whereas in the habitual standing position, this pressure is almost 2.5 times lower. The results of our study showed a significant increase in the value of the angle of trunk inclination when table tennis players adopted the ready position. This angle increased more than tenfold, hence the pressure on intervertebral discs probably increased, mainly in the lower part of the lumbar spine. The lower part of the lumbar spine (L4–S1) is characterized by greater mobility than its upper part, covering 95% of its entire range. On the other hand, in the places of the greatest mobility, with additional loads, there are exceptional possibilities of overloading and the appearance of symptoms of overload disease and pain.

The problem of overload, spinal pain or risk of injury resulting from faulty posture or morphological asymmetry seems to be common, especially in one-sided and monotypic sports. The main factors of various types of injuries in literature most often mentioned are those resulting from many hours of training and overloads and the specificity of the sport that is, multiple repetitions that overload specific parts of the body (*Saragiotto, Di Pierro & Lopes, 2014*).

The results of our survey confirmed that professional table tennis players experienced spinal pain, with nearly 32% of the respondents complaining about frequent and very frequent pain occurring mainly during the game or immediately after training. This frequency of pain occurrence is probably correlated with the value of angle of inclination of the thoracolumbar spine (β). As many as 36% of table tennis players had to stop training because of the pain. Slightly more than half of the respondents estimated the level of pain at the medium level (5–6 on the VAS scale) and three persons—at the level of 7–8. This demonstrates that the spine is probably heavily overloaded as a result of many hours of training of this sport, with the majority of the training time based on

adopting the position forcing the body to position body trunk at a significant forward inclination, and performing frequent and intensive torsional movements. Apart from the forced forward-leaning position, professional training in table tennis also forces the player to use only one upper limb while playing. Impact movements are very intense, often with the use of submaximal and maximal force, significantly involving the entire body. In order for the impact force to be maximal, the player must make a rotational movement, depending on the playing limb, from the maximum starting and ending ranges, combined with the transfer of body weight from one lower limb to another, but often with the feet on the ground being locked in one plane. Such forced repetitive movements put strain on the posterior lateral structures of the intervertebral discs, which may result in their damage. The results of our study showed that over 50% of the respondents complained about local pain, which may suggest that the overload to the perispinal structures is not yet at an advanced stage of the disorder.

The results of studies of other authors confirm the frequent occurrence of spinal pains in groups of people practicing various sports (rowers, dancers, fencers, gymnasts, athletes, figure skaters and shooters). They pointed out that this problem was mainly caused by high and substantial workout volumes (*Fett, Trompeter & Platen, 2017*; *Heneweer, Vanhees & Picavet, 2009*; *Trompeter, Fett & Platen, 2019*; *Wojtys et al., 2019*). Furthermore, they suggested that training should be monitored by experienced coaches to prevent back pain due to technical errors or too much strain exceeding the training capabilities of young athletes.

The training process in table tennis involves daily routines of many hours of exercise (usually from 4 to 6 h per day), which is observed even in young people at the age of 6 years due to the early specialization. The very young male and female athletes (Harimoto, Ito, Hirano) who are currently in the world's leading position (e.g., https://gossipgist.com/tomokazu-harimoto) are claimed to have started intensive training even at an earlier age. The analysis of the results in our study showed that table tennis players who declared high frequency of occurrence of back pain (Group 2) has been practicing longer than the others (Group 1). It can be assumed that in table tennis an increase of time of sport experience is accompanied by frequency of pain occurrence declared by participants. Taking all the above into account, it can be concluded that the risk of postural defects, spinal pain syndromes or morphological asymmetry exceeding the norm in table tennis may be high. It is worth noting that in the examined athletes, substantial asymmetry was found in the position of the scapulae and waist, while in other parameters, this asymmetry was at a moderate level.

An interesting observation also concerns the transverse and frontal planes. The table tennis players studied showed a greater pelvic torsion after adopting the ready position. This may be due to the specific body arrangement in the ready position, where the player positions the lower limbs asymmetrically, with the foot of the limb opposite to the playing upper limb moved forward. At the same time, it may be a signal that this pelvic asymmetry, which in literature is perceived as a consequence of the domination of one side of the body in sporting activities, may become permanent (*Bussey, 2010*). Significant asymmetry of pelvic rotation angle was also observed in the group of soccer players

(*Grabara, 2012*) and handball players (*Grabara, 2014*). It can be presumed that sport-specific training in asymmetrical sports can lead to asymmetry in the position of the body parts, which over time can be fixed in the habitual position.

The study also found an increase in asymmetry within the UK parameter (maximum deflection of spinous process line from the line C7–S1) in the frontal plane, which may indicate asymmetrical, unilateral bending of the spine in the ready position. Maintaining such a position for many hours can be conducive to various types of overload and asymmetrical muscle work. Therefore, the practical value of this study may be the observation that training programs should incorporate exercises that relieve the spine in the vertical line and exercises that improve symmetry of the work of the upper limbs, body trunk muscles and the pelvis. The results of our research indicated the need to supplement sports training with physiotherapy methods. These methods should reproduce the lordotic flexion of the spine in relief positions and include exercises to strengthen the postural muscles responsible for proper pelvic anterior tilt. Very important are also exercises which strengthen the muscles of the torso and upper limb of the non-dominant side and to shape the habit of correct body posture based on the symmetry of the shoulder and hip girdle. Limitation of our study could be some of errors that may appear during measurement, reported in the literature (*Mrozkowiak & Strzecha, 2012*). Another limitation of our work is a relatively small number of participants and a fairly large dispersion (variability) of their age. However, it is not easy to choose a study group consisting of female table tennis players who practice the sport professionally and with a sufficiently long training period. An insignificant asymmetry found in the frontal plane of the study participants (only in the case of the UK and KSM) was also surprising. Our previous research suggested the likelihood of a large asymmetry associated with practicing table tennis, especially in KLB, (Barczyk, Bańkosz, Derlich, 2012). Perhaps the participants of the present study are subjected to corrective and compensatory exercises in the direction that counteracts the asymmetry. The limitation of our study could be also interpretation of magnitude of asymmetry adopted in the research according to *Bibrowicz (1995)* which was originally designated to children

## CONCLUSIONS

This study demonstrated the dominance of kyphotic body posture in table tennis players., which can be caused by many hours of using the ready position during playing. After adopting this ready position, there are significant differences in the angles of anterior and posterior spinal curvatures compared to the habitual posture. This may be the cause of overloads and pain complaints reported by the study participants. Adopting the ready position is also associated with an increase in asymmetry in the position (transverse palnerotation) of the pelvis and spinous processes (frontal plane). Therefore, training programs should be extended with exercises that relieve the spine in the vertical line and exercises that improve symmetry of the work of the upper limbs, body trunk muscles and the pelvis. The need to entering compensation and correction programs to a training process confirms frequency of pain occurrence declared by participants which is accompanied by increase of thoracolumbar inclination and time of sport experience

### Funding
The authors received no funding for this work.

### Competing Interests
The authors declare that they have no competing interests.

### Author Contributions
- Ziemowit Bańkosz conceived and designed the experiments, performed the experiments, analyzed the data, prepared figures and/or tables, authored or reviewed drafts of the paper, and approved the final draft.
- Katarzyna Barczyk-Pawelec conceived and designed the experiments, performed the experiments, analyzed the data, prepared figures and/or tables, authored or reviewed drafts of the paper, and approved the final draft.

### Human Ethics
The following information was supplied relating to ethical approvals (i.e., approving body and any reference numbers):

The Senate's Research Bioethics Commission at the University School of Physical Education in Wrocław approved the study.

### Data Availability
Raw data is available as a Supplemental File.

### Supplemental Information
Supplemental information for this article can be found online at http://dx.doi.org/10.7717/peerj.9170#supplemental-information.

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
