# Peer review of "Habitual and ready positions in female table tennis players and their relation to the prevalence of back pain"

_PeerJ, doi:10.7717/peerj.9170_

## Round 0.1 · original submission · Major Revisions

Authors, thank you for your manuscript submission to PeerJ. I apologize for the delayed decision, but I had to source suitable reviewers. Please address the reviewer comments.

Thanks, A/Prof Mike Climstein

Reviewer 1 ·

Basic reporting

The article is written in English and use clear, unambiguous, technically correct text. The article is conform to professional standards of courtesy and expression.
The article include sufficient introduction and background. Literature is appropriately referenced.

The structure of the article is in acceptable format

Figures are relevant to the content of the article and appropriately described and labeled.

All appropriate raw data have are made in accordance with of Data Sharing policy.

The manuscript include all results relevant to the hypothesis.

Experimental design

The manuscript is original primary research within Aims and Scope of the journal.
The manuscript should clearly define the research question. The knowledge gap being investigated should are identified.
The investigation has high technical standard.
The research were conducted with the prevailing ethical standards in the field.
Research procedures described very clearly.

Validity of the findings

The presented results were based on objective research and bring practical solutions to sports training of table tenis.
The data and conclusions are based on available in an acceptable discipline-specific repository. The data are robust, statistically sound, and controlled.
The conclusions presented in the article contain the results, the following conclusions should be described in more detail from the three lines of text 338,339,340. The rest of the text from this section should be moved to the results and discussions section.

Additional comments

The manuscript presented is original contains, objective research, results are original and conclusions concurrently Apply the development of better training methods for famale playing table tenis.
However, it is necessary to rebuild the chapter of conclusions.
Results are not conclusions, so results should be removed from this chapter and the focus should be on the use of the results in practice.
The importance of the results obtained should be thoroughly discussed in terms of improving the efficiency of table tennis players' training and reducing pain felt during training.

Reviewer 2 ·

Basic reporting

This article writing conforms to the writing structure of the thesis.
Line 58, "(Fig. 1, Hudetz, 2005)" is not found in the References.
Lines 246-249, "The aim of the study was to …………… and to compare the body posture of the athletes studied with their non-athlete peers." No measurement data with non-athlete peers was shown in this article.

Experimental design

Lines 111-112, “The aim of the study was to evaluate the effect of body position during play on the change in the shape of anterior-posterior spinal curvatures and trunk asymmetry in table tennis players.” This study measured the difference between table tennis player’s ready position and habitual position. There must be a difference in body structure when posture is changed. The relationship between this difference and spinal pain should be explored.

Validity of the findings

This research concluded, “after adopting this position (table tennis ready position), there are significant differences in the angles of anterior and posterior spinal curvatures compared to the habitual posture. Adopting the ready position is also associated with an increase in asymmetry in the position of the pelvis and spinous processes.” Is this a table tennis player plays with one hand and the inevitable difference in the posture change?
Lines 55-58 , "The so-called ready position, in which the lower and upper limbs are flexed, the torso is significantly leaned forward, the rocket is kept in front of the body, the center of gravity of the body is shifted forward, the body weight is kept on the forefoot, etc. (Fig. 1, Hudetz, 2005)" This seems to indicate the possible results of this study. If this study could analyze the correlation between spinal pain and body posture of table tennis players, it would strengthen the value of this research.
In the conclusion section, “After adopting this position, there are significant differences in the angles of anterior and posterior spinal curvatures compared to the habitual posture. This may be the cause of overloads and pain complaints reported by the study participants.” Ready position is basically relaxed and not fixed for a long time. During the game or practice, players must change different ways to hit the ball, thus changing different body positions. From this point of view, it is difficult to determine the cause of pain complaints. Is it possible that the player's high-impact actions such as smash or loop drive during the game may be the cause of pain complaints? Therefore, if the relationship between spinal pain complaints and posture differences can be found, it would be more convincing. Pain complaints may also be caused by other improper postures or weekness of related muscle, not necessarily caused by the ready position.
Line 261, "hence the pressure on intervertebral discs increased significantly." This is only speculation based on the results of Nachemson (1987), and had not been confirmed in this study.

Additional comments

In summary, this study completed measuring the difference of table tennis player’s ready position and habitual position, and the investigation of spinal pain complaints. Although kyphotic body posture and asymmetry in the ready position was found in table tennis players. If the study could figure out the association between spinal pain and body posture, it could improve the readability of this study. Lines 109-110, “Can overload occur in certain parts of the spine and can the asymmetry deepen as a response of adopting this position?” The results of the study did not answer the hypothetical question raised by the authors. Therefore, publication is not recommended.

Reviewer 3 ·

Basic reporting

no comment

Experimental design

1. The title in my opinion doesn’t convey the essence of the topic, it need to be overthink and authors should try to not to use suggestive words as “asymetry” or “differency” (my recommendation eg.: Effect of playing table tennis position on posture female players).
2. What were the sampling methods? Please complete into materials and methods/ participants part. probability or non-probability (Simple random sampling, Stratified random sampling, Systematic random sampling, Multistage random sampling, Cluster sampling/Judgement Sampling, Quota Sampling, Convenience Sampling, Extensive Sampling). There is lack of the information how the authors selected the study group

Validity of the findings

1. Conclusions need to be re-edited, there is not necessarily to give any explanation; my suggestion: This study demonstrated the dominance of kyphotic body posture in table tennis players., which can be caused by many hours of using the ready position during playing. After adopting this ready position, there are significant differences in the angles of anterior and posterior spinal curvatures compared to the habitual posture. This may be the cause of overloads and pain complaints reported by the study participants. Adopting the ready position is also associated with an increase in asymmetry in the position (transverse palnerotation) of the pelvis and spinous processes (frontal plane). Therefore, Training programs should be extended with exercises that relieve the spine in the vertical line and exercises that improve symmetry of the work of the upper limbs, body trunk muscles and the pelvis.
2. line 145: who was doing the examination, marked points? it was the same person? was the person a physiotherapist or experienced researcher?
3. line 146: in methodology of the mora there is necessary to mark also a peak of lordosis and kyphosis. does the authors marked this points also?
4. line 178: norms by Bibrowicz (Bibrowicz 1995) are for adults or children, men or women? Please explain this or assign this fact as a limitation of the study.
5. line 189: the player carried the racket during keeping the ready position or not? if not explain why?
6. line 222: Explain what is the value of compensation index and on what is based this index?
7. line 229: “moderate level” Please explain on which form was based this statement?

Additional comments

1. There is different English name of the school on the webside: Faculty instead department; and there is a typing error (Department of Sports, Universisty School of Physical Education in Wrocław, Wrocław, Poland).
2. Text is not justified.
3. Single typing or editorial errors throughout the text, please trace the text: line 174, line 137,line 176,
4. line 29: add (averrage, sd) age, body mass, time of study population study population.
5. line 30: completed a questionnaire. add a :author's own” questionnaire
6. line 51: delete unnecessary “- -”
7. line 68: correct “(excessive kyphosis, lordosis, scoliosis)”; my suggestion: Postural defects (excessive spine curvatures, scoliosis)
8. line 70: even internal organs.. add “even internal organs functioning.”
9. line 77: “muscle tears” correct into “muscle strain”
10. line 103 and 109-111: I recommend to remove this sentence or transfer to discussion
11. line 123: the same as in line 30
12. line 137: “that maps the anteroposterior curvature” add a “spine”
13. Table 1: there is no explanation for the “*” symbol. Center the text in the table.
14. line 246-253: This is unnecessary repetition from introduction part.
15. Authors should indicate the limitations and strengths of the study (there are some more of them except of indicated eg. Mora measurement error, sampling method; indicate the postural problems in group of table tennis players)

---

## Round 0.2 · accepted · Accept

Dr Mankosz,

Congratulations - I am pleased to inform you that your manuscript has been accepted for publication by the Reviewers who have accepted your amended manuscript.

My apologies again for the delay in processing your manuscript however none of the Reviewers you recommended would agree to review your manuscript. Congratulations again. A/Prof Mike Climstein

Reviewer 1 ·

Basic reporting

The authors have followed all of my corrections. The work meets all criteria for publication.

Experimental design

The authors have followed all of my corrections. The work meets all criteria for publication.

Validity of the findings

The authors have followed all of my corrections. The work meets all criteria for publication.

Additional comments

The authors have followed all of my corrections. The work meets all criteria for publication.

Reviewer 3 ·

Basic reporting

No Comment.

Experimental design

No Comment.

Validity of the findings

No Comment.

Additional comments

Thank you for your solid corrections and explanations! It was a pleasure to do a review your paper.